# Porphyrin-Based Sorbents for the Enrichment and Removal of Metal Ions

**DOI:** 10.3390/molecules30102238

**Published:** 2025-05-21

**Authors:** Krystyna Pyrzynska, Krzysztof Kilian

**Affiliations:** 1Faculty of Chemistry, University of Warsaw, Pasteur 1, 02-093 Warsaw, Poland; 2Heavy Ion Laboratory, University of Warsaw, Pasteur 5A, 02-093 Warsaw, Poland; kilian@slcj.uw.edu.pl

**Keywords:** porphyrins, solid-phase extraction, porous organic polymers, carbon nanostructures, metal–organic frameworks, metal ions

## Abstract

Porphyrins and their derivatives are excellent materials with specific physical and photochemical properties in medical, chemical, and technological applications. In chemistry, their properties are applied to create new functional materials with specific characteristics, such as porphyrin-based sorbents combined with porous organic polymers, silica, carbon nanostructures, or metal–organic frameworks. This review covers the applications of porphyrins and metalloporphyrins in preparing and using sorbents for metal ion enrichment and their separation. Uncommon applications that utilize specific properties of porphyrins, such as light-enhanced processes and redox properties for selective sorption and photocatalytic conversion of metal ions, are also discussed. These applications suggest new fields of use, such as the removal or recycling of metals from electronic waste or the selective elimination of heavy metals from the environment.

## 1. Introduction

The level of heavy metals in wastewater has been increasing due to industrial growth and human activities. Discharging contaminated wastewater directly exposes aquatic life to the dangers, and bioaccumulation of these metals in the food chain can harm humans and animals [1,2]. Hg, Pb, Cd, As, Cr, Cu, and Zn are major heavy metal pollutants from industrial effluents. The exposure to these metals is associated with liver and kidney damage, increased cancer risk, reduction in hemoglobin formation, hypertension, cardiovascular issues, and nervous diseases [3,4].

Recently, strong focus has also been directed toward the occurrence of heavy metal speciation in wastewater, as the potential risk, toxicity to organisms, and bioavailability are strongly dependent on the chemical species of metals rather than their total content [5,6]. For example, Cr(VI), which is widely used in the electronics industry, tanning, and metal electroplating, can cause a range of negative health effects, including carcinogenicity, genotoxicity, and dermatosis. It can also damage the liver, kidneys, and gastrointestinal tract, necessitating strict control, limitation, and removal of excess chromium(VI) from the environment [7]. In contrast, Cr(III) is an important micronutrient involved in carbohydrate, lipid, and protein metabolism.

Various remediation approaches with differing degrees of efficiency have been developed for metal ion removal, including chemical precipitation, advanced oxidation, membrane filtration, adsorption, solvent extraction, electrochemical treatment, coagulation, and flocculation [8,9,10]. New methods include membrane bioreactors, electrocoagulation, and bioremediation by microorganisms. Each of these methods has certain advantages and drawbacks, depending on the specific metal, water quality, and intended results. These methods have been discussed and compared in recent reviews [11,12,13]. Among various methods, adsorption is the most widely used due to its simplicity, low cost, and the availability of diverse adsorbents [14,15]. Developing efficient, eco-friendly, and cost-effective sorbents remains a key research focus in water contaminant removal [16,17].

In analytical chemistry, the direct determination of trace concentrations of analytes in the presence of relatively complex matrices is often problematic. Thus, sorption methods are very helpful for the preconcentration, separation, and speciation analysis of metal ions [18,19]. This preliminary step in an analytical procedure is often necessary to increase the sensitivity and selectivity of the applied analytical methods. Solid-phase extraction (SPE) and dispersive micro-solid-phase extraction (DMSPE) are commonly used in the sample preparation step. Beyond conventional sorbents, numerous novel materials are being developed and modified through functionalization, doping, or composite fabrication to improve their sorption properties and selectivity [20,21,22]. The incorporation of various functional groups enhances interactions with metal ions via ion exchange, complexation, and electrostatic attraction.

Porphyrins, distinguished by their unique macrocyclic structure consisting of four pyrrole rings, are excellent complexing agents for metal ions. Their intrinsic cavity and chelating nitrogen atoms provide a strong binding effect [23]. The tetrapyrrole structure of porphyrins can be modified at peripheral positions with various side chains or substituents, resulting in a wide range of derivatives. Metalloporphyrins, owing to their distinctive chemical properties, photoelectrochemical characteristics, and biological significance, have numerous applications across diverse fields. In biomedicine, they have been utilized not only as tumor diagnostic agents [24] but also as photosensitizers in photodynamic cancer therapy [25], contrast agents in magnetic resonance imaging [26], antimicrobial agents in photodynamic chemotherapy [27], and enzyme models in bioinorganic chemistry [28]. In analytical chemistry, porphyrins serve as sensors and dyes, facilitating the detection of various analytes through fluorescence and absorbance measurements [29,30,31].

Porphyrins and their derivatives serve as excellent building blocks for various porous materials, including polymers, silica, metal–organic frameworks (MOFs), covalent organic frameworks (COFs), and magnetic nanocomposites [21]. These materials benefit from the advantageous properties of porphyrins, such as their absorption capabilities, redox activity, coordination chemistry, as well as the high surface area and porosity of the supporting structures. Porphyrin-based porous materials are being investigated for a wide range of applications, including gas and energy storage [32,33], catalysis and photocatalysis [34,35], separation processes [36], sensing [37], and wastewater remediation [38].

The development and application of porphyrin-based materials have significantly increased in recent years, leading to a gradual rise in the number of related review papers. Some of these reviews focus exclusively on porphyrin-based porous organic polymers [39,40,41] or covalent organic frameworks [42,43], while others explore the application of porphyrin-based materials in sensing and the removal of organic pollutants [29,31] or in biomedical fields [44,45].

A literature search in multidisciplinary citation databases covering the past 10 years identified available primary studies (Figure 1), which account for more than 60% of all published papers in this field, highlighting its dynamic development. In contrast, an earlier work from 2000 [46] did not mention the application of porphyrins for metal sorption, likely because the first publication on porphyrin cross-linking dates back only to 1994 [47], and significant development in this area began in the 2010s.

This article was conceived as a critical review aiming to identify studies that report experimental results on the synthesis, characterization, and performance of porphyrin-based sorbents for metal ions, while also highlighting the strengths and weaknesses of these materials in practical applications. This review specifically focuses on porphyrin-based materials used as sorbents for metal ion removal and enrichment. Following a brief overview of their synthesis, the discussion centers on their applications in metal enrichment and separation.

Note: The removal of metal ions from a solution by solid materials includes their bonding to its surface and their penetration into a network. Thus, according to the suggestions by Pourret et al. [48], we use the term sorption, which combines the processes of adsorption and absorption.

## 2. Porphyrin-Based Sorbents

The properties of porphyrin-based sorbents strongly depend on the characteristics of their skeletal structure and the composition of porphyrin ligands. Materials with a high specific surface area and developed pore structure favor the enrichment of metal ions. Their high sorption capacity is particularly beneficial in applications where target metal ions are present in low concentrations, together with a complex mixture of competing substances. Additionally, the sorbent’s ability to be regenerated and reused multiple times without significant loss of its performance should also be taken into consideration. 

### 2.1. Characterization of Porphyrin-Based Materials

The synthesized porphyrin-based materials have been characterized using microscopic and spectroscopic techniques [49]. These techniques provide valuable insights into size, shape, chemical composition, surface properties, crystal structure, and stability. Very often, multiple methods are used for comprehensive material analysis to show the full potential of these materials in diverse fields. To confirm the presence of porphyrinic moieties in the prepared materials, Fourier-transform infrared (FTIR) and ^13^C NMR spectroscopy are employed. These methods enable the identification of capping ligands and the analysis of surface composition. X-ray photoelectron spectroscopy (XPS) is also highly sensitive to surface modifications. Scanning electron microscopy (SEM), transmission electron microscopy (TEM), and dynamic light scattering (DLS) are commonly used to examine particle size, morphology, agglomeration, and distribution. The nitrogen adsorption–desorption isotherms are applied to determine the specific surface area and porous structure. Surface charge is another parameter that evaluates the behavior of materials, particularly in an aqueous environment and is usually evaluated by estimating the pH of zero point of charge (pH_ZPC_). Additionally, thermogravimetric analysis is used to study the thermal stability of synthesized sorbents.

### 2.2. Porphyrin–Silica Materials

Mesoporous silicas are among the most commonly used supports for the preparation of porphyrin-based adsorbents due to their good mechanical, thermal, and chemical stability. The selectivity of these materials can be tailored by incorporating different organic bridging groups into the organosilica structure and by selecting suitable porphyrin components. The first step in the synthesis of such adsorbents involves the reaction between a silylating agent, typically 3-aminopropyltrimethoxysilane (APTMS), and the silanol groups present on the silica surface. The resulting functional groups, such as -NH_2_, subsequently react with the chosen porphyrin ligand in an anhydrous toluene/ethanol mixture under reflux and a nitrogen atmosphere [50,51,52,53] (Figure 2).

The efficiency of hybrid material obtained by immobilizing 5,10,15,20-tetrakis (pentafluorophenyl)–porphyrin (H_2_TF_5_PP) on a silica surface was evaluated for removal of Pb(II), Cu(II), Cd(II), and Zn(II) from water [50]. The optimum retention for lead, copper, and zinc ions on SiO_2_@TF_5_P-porphyrin was observed at pH values between 6 and 7, while for cadmium it was in the range of 5 to 7. The adsorption capacity followed the sequence Pb(II) > Cu(II) > Cd(II) > Zn(II), with respective sorption capacities of 187.36, 125.16, 82.44, and 56.23 mg/g. The different selectivities were explained by considering the activation energy of the rate-limiting step for each metal ion, influenced by the size of the metal ion and the deprotonation step of the two N–H groups when the metal is incorporated into the macrocyclic cavity. The sorbent material remained unaffected after five cycles of use. SiO_2_@TF_5_P–porphyrin sorbent exhibited high affinity for Pb(II), even in the mixture containing competing metal ions. Practical tests using river water samples demonstrated that the presence of other ions (Na^+^, K^+^, Ca^2^^+^, Mg^2^^+^, NH_4_^+^, SO_4_^2^^−^, NO_3_^−^, HCO_3_^−^, PO_4_^3^^−^) did not interfere with the sorption of lead ions, and this material remained unaffected after five cycles of use.

The same group of researchers tested two 5,10,15,20-tetrakis-(pentafluorophenyl)porphyrin derivatives, with three (Si_3_PyS) and four (Si_4_PyS) mercaptopyridyl substituents, grafted onto silica gel functionalized with APTMS [51]. The affinity of these materials for heavy metal ions at pH 6 decreased in the sequence Cu(II) > Pb(II) > Zn(II) > Cd(II). Both materials showed high sorption capacities, ranging from 176.32 mg/g (Si_3_PyS) and 184.16 mg/g (Si_4_PyS) for Cu(II) to 73.84 mg/g and 84.54 mg/g for Cd(II), respectively. The authors suggest that the high adsorption of Cu(II) may be due to its smaller ionic radius. However, compared with the earlier results [50], it seems that the presence of mercaptopyridyl substituents also plays an important role.

Radi et al. [53] studied the adsorption efficiency of SiO_2_@NTPP (3-aminopropyl-silica functionalized with 2-formyl-5,10,15,20-tetraphenylporphyrin) towards Pb(II), Zn(II), Cu(II), and Cd(II). The retention was not significant at a low pH, as the porphyrin core was fully protonated. At pH 5–7, protonation decreased, and adsorption dramatically increased, but varied for different ions: the optimum pH for maximum adsorption of Pb(II) and Cd(II) was at pH ≥ 5, while for Zn(II) and Cu(II), the maximum adsorption was at pH ≥ 6. The adsorption capacity followed the sequence Pb(II) > Zn(II) > Cd(II) > Cu(II) and reached 53.20, 32.16, 23.26, and 19.07 mg/g, respectively. SiO_2_@NTPP showed resistance to repetitive sorption and desorption for at least five cycles of sorbent regeneration and maintained stable recovery of around 94–98% in each cycle.

Yu et al. synthesized a porphyrin-based magnetic nanocomposite by modifying SiO_2_-coated Fe_3_O_4_ nanoparticles with 5,10,15,20-tetrakis(4-carboxyphenyl)porphyrin (TCPP) [54]. The resulting nanocomposite exhibited a core-shell structure, with Fe_3_O_4_ encapsulated within a silica shell. TCPP was grafted onto the SiO_2_ surface, forming an additional shell approximately 30 nm thick. The sorbent exhibited favorable kinetics for Pb(II) sorption, reaching equilibrium in 5 min, regardless of the initial lead concentration. However, Pb(II) concentration influenced the adsorption mechanism. When the initial concentration was lower than 19.2 mg/L, the monolayer adsorption process fit well with the Langmuir isotherm and was based on the coordination of Pb(II) and TCPP. For higher Pb(II) concentrations, in the range of 19.2–512 mg/L, a multilayer adsorption, internal crystallization, and limited solubility contributed to the final properties, with the influence of crystallization increasing with concentration. The sorption capacity was calculated as 798.34 mg/g, and Pb(II) removal reached 84.3% at an initial Pb(II) concentration of 16 mg/L. DFT calculations showed that Pb(II) was preferentially coordinated with the four nitrogen atoms in the tetrapyrrolic core rather than with Si-O-Si, -NH, or C=O groups, indicating that the porphyrin core promoted stronger removal of Pb(II) than other materials with carboxylic, amino, or hydroxylic functional groups.

### 2.3. Porphyrin-Immobilized Resins

Some organic complexing reagents are sorbed onto the hydrophobic surface of conventional anion-exchange resins and non-ionic sorbents that lack functional groups. This process results in the formation of so-called “ligand-immobilized resins” or “modified resins”, which can interact with various metal ions through complex formation. In general, the immobilization of ligands onto a polymeric matrix involves ion exchange and/or adsorption mechanisms [55].

When using anion exchangers, retention via the ion exchange mechanism occurs through the replacement of chloride, nitrite, or acetate ions bound to the quaternary ammonium groups of the resin with the negatively charged groups of the ligand (such as sulfonate or carboxylate), which do not interfere with chelation. In addition, porphyrin retention may result from physical interactions, specifically π-π dispersion forces arising from the aromatic structures of both the resin and the ligand. Polymers containing functional groups capable of binding metal ions offer several advantages, including ease of operation, regeneration, and reusability.

TCPP was immobilized on microporous conventional polymeric anion-exchange resins Amberlite IRA-401 and Amberlite IRA-904, as well as on the non-ionic adsorbent Amberlite XAD-2 [56]. The adsorption capacities of the porphyrin ligand at pH 9 were 0.28, 0.23, and 0.14 mmol/g, respectively. TCPP retention on Amberlite XAD-2 occurs solely through π-π interactions between the aromatic structure of the porphyrin and the polymeric matrix, resulting in a lower adsorption capacity. In contrast, an additional ion exchange mechanism is involved in the case of Amberlite IRA resins, contributing to their higher adsorption capacities.

### 2.4. Porphyrin-Based Porous Organic Polymers

Porphyrin-based porous organic polymers (P-POPs) are highly polymeric backbones composed of one or more structured building blocks, with their structure determined by the composition and connectivity of functional units [40,57]. The diversity of functional units, linkage bonds, and connection methods contributes to the structural variability of these materials, enabling convenient post-functionalization and extended π-conjugation.

There are many types of P-POPs, such as covalent organic polymers (COPs), conjugated microporous polymers (CMPs) with low skeleton density, and covalent organic frameworks (COFs), whose skeleton structures are formed by light elements (C, B, O, Si, N) that connect organic matter through strong covalent bonds. COFs, in contrast to POPs, are typically crystalline. Other types include hyper-cross-linked polymers and porous aromatic frameworks, which consist of aromatic units linked by strong C–C bonds and can remain stable under harsh chemical conditions [40].

Two main methods are used to synthesize porphyrin-based porous organic polymers [40,58,59]. The first method is a one-pot synthesis using the Alder–Longo approach, in which porphyrin macrocycles are directly formed through the cyclic tetramerization of pyrrole with monomers containing multiple aldehyde groups during the polymerization process (Figure 3). In addition to the porphyrin core, various coordinating moieties can be introduced at the peripheral positions of the porphyrin, enabling diverse structures and applications. The second method involves the direct polymerization of porphyrins functionalized with specific groups, along with other building blocks, through classical catalytic reactions or electrooxidation. The functional groups in porphyrin monomers (such as aldehyde, amino, fluorine, terminal alkyne, hydroxyl, and carboxyl) should be positioned at the meso site, ensuring compatibility within the reversible covalent bonding synthesis system. Various covalent bond-forming reactions employed in the synthesis of porphyrin-based COFs have been recently discussed [39,42,43].

Nguyen et al. synthesized P-POPs via a coupling reaction between TNPPH_2_ (5,10,15,20-tetrakis(4-nitrophenyl)-21H,23H-porphyrin) monomer and a diamine–arene linker [60]. This porphyrin–phenazine polymer exhibited a high capacity for Au(III) ion retention, reaching 1315 mg/g at pH 2. The adsorptive-reductive mechanism underlying this sorbent’s uptake activity was postulated based on X-ray photoelectron spectroscopy (XPS) and elemental mapping. Gold particles are formed within the porous network due to the reductive action of nitrogen centers. The retention experiments were also conducted in the dark and under constant irradiation. A diminished capacity of 1188 mg/g was observed without a light source, while an enhancement to 1354 mg/g was noted with continuous illumination from a halogen light. The authors explained this increase by the photoinduction of the porphyrin core for the reduction of gold ions. However, it is debatable whether this capacity increase after light irradiation is high enough to be economically viable. The proposed porous porphyrin–phenazine network also exhibited a significant affinity for Pt, Pd, Ag, and Cu ions. The efficiency of gold desorption from TNPPH_2_ was investigated using different eluents, including HCl-HNO_3_ mixtures, thiosulfate–sulfite, and thiourea solutions [61]. Thiourea solution enabled the release of approximately 97% of the gold, acid mixtures achieved ~75%, while thiosulfate–sulfite recovered less than 5%. Under optimal conditions (0.1 M thiourea in 0.1 M sulfuric acid at 50 °C for 6 h), gold desorption efficiency reached up to 97%. The sorbent’s reusability was confirmed over five consecutive regeneration cycles, maintaining 94% desorption efficiency in the final stage.

The cationic porphyrin-based POP with imidazolium functional groups also exhibits an exceptionally high capacity for Au(III) (q_max_ = 1543 mg/g) [62]. The retention mechanism was attributed to electrostatic interactions between Imi-P-POPs-Br and AuCl_4_^−^, strong chelation with nitrogen functional groups, and reductive immobilization in the form of elemental gold. Gold retention remained highly selective even in the presence of multiple competing metal ions, while chloride, carbonate, sulfate, and nitrate anions had a minimal impact on sorption. The 2D/3D porphyrin-based porous polyaniline derivatives fabricated by chemical oxidation polymerization also enabled highly efficient adsorption of Au(III) with q_max_ of 1152 mg/g [63].

Tetrakis(4-aminophenyl)porphyrin (TAPP), functionalized with the additional hydroxyl groups, was used as monomers for COPs synthesis and applied for Cd(II) removal and detection [64]. The XPS spectrum indicated the participation of abundant hydroxyl groups in Cd(II) coordination. The adsorption capacity at pH 8 was calculated to be 166 mg/g. The proposed sorbent could be used not only for the removal of toxic cadmium ions but also as a fluorescent probe for their detection in solution and quantitative determination at 538 nm (λ_ex_ 468 nm) with a detection limit of 1.5 µg/L. To simplify the Cd(II) removal process, sponge-supported monolithic materials were prepared via in situ growth of TAPP-COF within the pore channels of a heat-resistant polyurethane sponge. Dopamine was used in the pretreatment step to increase the number of amino groups on the sponge surface and pores, facilitating COF growth. During this process, the pore sizes of the sponge were enlarged due to the increase in material rigidity, and cadmium ions could move freely within the pores and interact with the COFs. As a result, this new nanomaterial exhibited faster sorption kinetics and greater resistance to competitive metal ions.

TAPP and 1,3,5-triformylphloroglucinol (Tp) were used as monomers for COF synthesis, exhibiting a high adsorption capacity for Cd(II) due to their multiple coordination sites [65]. With increasing cadmium concentration, the absorbance at 434 nm (characteristic of the monomers) decreased, while the absorbance of a new band at 462 nm increased. This redshift of the Soret band of this material was proposed to serve as an optical probe for the rapid (5 min) ratiometric determination of cadmium in the concentration range of 0.02–9 mg/L, with a detection limit of 0.016 mg/L. In addition to this application, the synthesized COF also functioned as a sorbent for Cd(II) removal from contaminated samples via centrifugation. The authors proposed its use in the form of a hybrid membrane, prepared via the in situ growth of a TAPP-Tp-COF network on a carbon fiber surface, followed by mechanical pressing. The preparation scheme of the CF@TAPP-Tp-COF membrane and the most energetically favorable Cd(II) coordination model are presented in Figure 4. This CF@TAPP-Tp membrane exhibited an adsorption capacity of 75.2 mg/g for cadmium and could be regenerated using a 0.1 M HNO_3_ solution.

Li et al. [66] proposed a cationic organic network, synthetized in a quaternization reaction between 1,4-bis(bromomethyl)benzene and 5,10,15,20-tetra(4-pyridyl) porphyrin (TPP). The material with exchangeable Br^−^ ions was a solid powder, resistant to common solvents, such as water, N, N-dimethylformamide, tetrahydrofuran, or methanol. Despite the lower surface BET area (26 m^2^/g) than other similar porous materials, it showed good sorption properties for Cr(VI) ions via an anion exchange process. When 2 mg of the synthesized network was immersed in K_2_Cr_2_O_7_ solution (6 mL, 100 mg/L), chromium concentration decreased by 68.6% in less than 3 min, and quantitative sorption was obtained after about 60 min. Adsorption kinetics indicated a pseudo-second order with a kinetic constant of 5.06 × 10^−3^ g/mg·min. The determined maximum capacity for Cr_2_O_7_^2−^ was 293 mg/g.

Lone et al. [67] studied the sorption of chromium(VI) on a 5,10,15,20-tetrakis(4-methoxy phenyl)porphyrin (4MPP) conjugated microporous framework complexed with cobalt(II) ions with its pH_PZC_ value of 6.35. The sorption capacity for this framework was 339.17 mg/g at pH 2. The proposed sorption mechanism involved reduction, complexation, and electrostatic interactions induced by protonated nitrogen and oxygen in the functional groups of porphyrin. The material maintains 70% of its sorption efficiency after four adsorption–desorption cycles. A similar covalent organic framework for Cr(VI) removal was synthesized based on TCPP and complexed with Zn(II) [68]. The obtained materials exhibited high sorption capacities: 246.76 mg/g for TCPP-COF and 273.09 mg/g for Zn-TCPP-COF. The photocatalytic properties, specifically the reduction of Cr(VI) to Cr(III), were more pronounced for the metalloporphyrin complex, leading to enhanced Cr(VI) sorption. It was explained through a synergistic mechanism: Zn ions act as a trap for photoelectrons during photoexcitation, enhancing photocatalytic efficiency. Simultaneously, adsorbed Cr(VI) is reduced to Cr(III), improving charge balance and decreasing electrostatic repulsion between Cr(III) ions and active sites. This, in turn, increases Cr(VI) adsorption via electrostatic interactions. Adsorption kinetics were evaluated using both pseudo-first-order and pseudo-second-order models, and similar correlation coefficients were calculated. Consequently, it was concluded that the sorption process follows a mixed mechanism involving both physical and chemical sorption.

**Figure 4 molecules-30-02238-f004:**
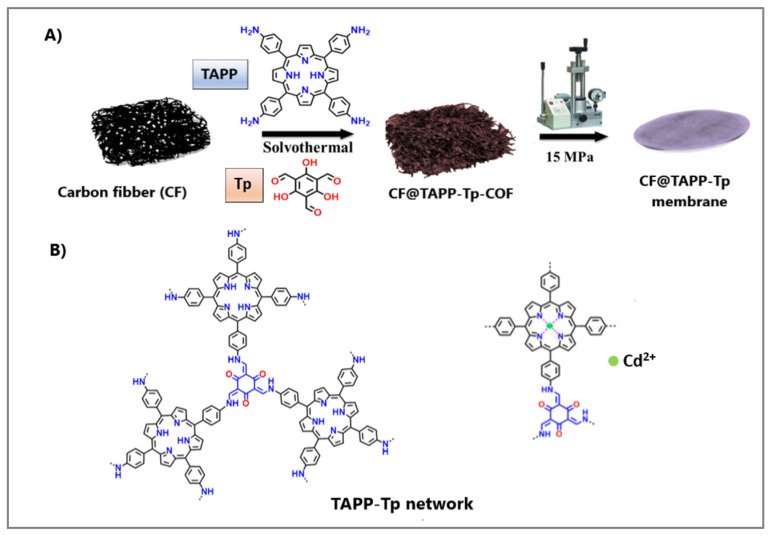
The scheme for the preparation of CF@TAPP-Tp-COF membrane (**A**) and the coordination model for cadmium ions (**B**). Reproduced with permission from reference [69]. Copyright Elsevier 2022.

A new approach for chromium preconcentration and detection, integrating adsorption and photochemical properties, was presented in [70]. The bifunctional material was synthesized by combining photosensitive 1,5,10,15,20-tetra(4-aminophenyl)-21H,23H-porphine (TAPP) with an aromatic tri-aldehyde [1,3,5-tris(3-(4-formylbenzyl)-1H-imidazol-1-yl)benzene bromide]. This design enables a mechanism in which a large number of cationic imidazole groups electrostatically interact with Cr(VI)-containing anions. Simultaneously, the protonated amine groups enhance the material’s electropositivity, further strengthening electrostatic interactions with Cr(VI) anions, providing additional binding sites, and improving the material’s performance. As a result, the COF exhibits effective adsorption with a maximum capacity of 373.14 mg/g. Moreover, the material demonstrates properties similar to those of a light-induced oxidase, allowing its use as a sensor in colorimetric measurements. The obtained results showed a linear detection range from 0.5 μM to 220 μM, with a detection limit (LOD) calculated at 41 nM. The method exhibits high resistance to interference from cations and anions typically found in natural waters. Its practical application was verified through the purification of simulated wastewater containing chromium at a concentration of 5 mg/L in the presence of other pollutants (100 mg/L solutions of Hg, Ni, Cd, and Co). Under these conditions, 99% of the chromium load was removed, producing drinking water of suitable quality.

Hg(II) ions can form strong coordination bonds with functional groups containing sulfur, nitrogen, and oxygen atoms. Thus, thiophene structures were incorporated into the porphyrin-based POPs, synthesized via the Adler–Longo reaction using aldehyde monomers (2,5-thiophenedicarboxaldehyde or thieno[3,2-b]thiophene-2,5-dicarboxaldehyde) and pyrrole [71]. The resulting thiophene-based porphyrin polymers (PSFs) contained 7.0% sulfur, while PS2F, constructed from two thieno[3,2-b]thiophene-2,5-dicarboxaldehyde units, had a sulfur content of 13.2%. The synergy between nitrogen in pyrrole and sulfur in thiophene structures resulted in a remarkably high adsorption capacity, reaching 1049 mg/g. Although the POP based on native porphyrin exhibited the highest BET surface area (648 m^2^/g), the increased sulfur content in PS2F played a more significant role in Hg(II) sorption. XPS spectra revealed strong interactions between Hg(II) and nitrogen atoms in the porphyrin core and sulfur atoms in thiophene. The thiophene-functionalized porphyrin polymers demonstrated high capacity (1049 mg/g) and favorable sorption kinetics—90% of Hg(II) was retained within 5 min and 99.9% within 30 min. They also exhibited excellent selectivity, with Hg(II) sorption being more than six times higher than for other competing cations. The polymers displayed good reusability, maintaining over 90% sorption efficiency after five cycles. Zare et al. proposed a porous organic polymer that contains a porphyrin ring and oxygen atoms, acting as coordinating sites towards Hg(II) [69]. The porous nature of this polymer, with a BET surface area of ~80 m^2^/g, was identified as a crucial feature for efficient adsorption performance, reaching the sorption capacity of 384.6 mg/g. A small volume (0.7 mL) of 1 M HCl solution effectively desorbed Hg(II) ions, and a low limit of detection (2.1 ng/L) was achieved.

### 2.5. Porphyrin-Based Meta–Organic Frameworks

Metal–organic frameworks (MOFs), also known as coordinated organic frameworks, are constructed by coordinating metal ions (or their clusters) with organic ligands [72]. In addition to their large surface area, high porosity, and structural flexibility, MOFs exhibit high mechanical strength and thermal stability but have limited chemical stability [73]. The incorporation of porphyrin ligands results in porphyrinic MOFs. In contrast, the immobilization of porphyrin molecules into MOFs via surface modification or pore encapsulation leads to porphyrin@MOFs, which are classified as porphyrin-based MOFs (P-MOFs) [29,74]. P-MOFs have garnered significant attention due to their potential applications in biomedicine [75], photocatalysis [76], environmental pollutant sensing [29], and the removal of heavy metal ions from wastewater [77]. Notably, in the design of new photocatalytic materials, the photocatalytic activity of porphyrin-based MOFs can be regulated and optimized by selecting an appropriate porphyrin structure, designing the porphyrin molecule, and incorporating mixed ligands into the MOF framework [78]. For example, a zirconium TCPP porphyrin-based MOF was synthesized for the detection of cadmium ions [79]. It exhibits fluorescence quenching in the presence of Cd(II) at concentrations as low as 0.3 µg/L, which is well below EPA guidelines, demonstrating its effectiveness as a chemosensor for water contamination. Zhang et al. constructed a similar porphyrin-based zirconium MOF using the solvothermal method for antimony adsorption [42]. Sb(V) was initially adsorbed onto the Zr cluster nodes, then distributed along the skeleton through hydrogen bonds, and finally anchored at the N-coordination fixed sites in the porphyrin ligand.

A Zr-based porphyrin MOF synthesized using TCPP exhibited an exceptionally high Au(III) recovery capacity of 2613 mg/g under visible light irradiation [80]. However, in the absence of light, the capacity dropped to 885 mg/g. A light-triggered adsorption-reduction mechanism was proposed based on data from various microscopic and spectroscopic techniques. Nevertheless, kinetic analysis indicated that the adsorption process preceded the redox reaction. Introducing acetate further enhanced network stability and increased gold recovery capacity to 4946 mg/g. The performance of Zr-TCPP-MOF in gold recovery was validated using real leaching solutions from e-waste. Despite potential interferences from coexisting metals and the acidity of the leachate, more than 99.9% of Au in the solution was successfully recovered, while the recovery of other metals remained below 6%.

Ren et al. reported the synthesis of (Fe–P)n-MOF from TCPP and FeCl_3_ and investigated its performance as a sorbent for recovering Au(III) and Au(I) ions [81]. In their experimental studies, these ions were present as AuCl_4_^−^ and Au(S_2_O_3_)_2_^3^^−^, under conditions mimicking thiosulfate and chlorate leachates (Figure 5). Sorption reached saturation within approximately 5 min, following the Freundlich isothermal adsorption model. After sorption, gold complex ions were partially reduced by free electrons released from the porphyrin ring, leading to the formation of gold nanoparticles that aggregated on the sorbent surface. The maximum adsorption capacities were 2026 mg/g for Au(I) and 2396 mg/g for Au(III). The superior ability to remove Au(III) was attributed to its easier reduction on the MOF compared to Au(I).

A similar MOF framework was studied by Zhang et al. [82] for Sb(V) preconcentration. Sb(V) was initially adsorbed onto the Zr cluster nodes, then distributed along the skeleton through hydrogen bonds, and finally anchored at the N-coordination fixed sites in the porphyrin ligand. Sb(V) was adsorbed in Zr clusters, then transported via hydrogen bonds to the porphyrin ligand and finally fixed by nitrogen atoms. The sorbent was then converted to Sb nanoparticles embedded in porous carbon and used for the construction of alkaline batteries. The influence of pH and foreign ions was similar to that in [79], with maximum adsorption at pH 3 and a significant inhibitory effect of SO_4_^2^^−^ ions.

A zirconium-TCPP-based organic framework was also studied for Sb(III) sorption performance in aqueous solutions [83]. The sorbent was formed by linking TCPP ligands to stable Zr_6_ clusters, showing resistance to high temperatures and concentrated acids. Removal of Sb(III) was studied in a pH range of 2–10, showing a combination of the influence of the adsorbent’s zeta potential (pH_ZPC_ = 9) and the properties of analyte species in solution. The maximum sorption capacity reached two peaks at pH = 2 and pH = 10, with two useful ranges at pH 2–4 and pH 8–10. In both cases, Sb(III) sorption was rapid, achieving more than 90% efficiency within 9 min, with a maximum value of 175.17 mg/g. This framework was applied to the treatment of simulated natural water spiked with Sb(III). The removal efficiency reached 80%, reducing the antimony content below the limits for drinking water. However, a significant reduction was observed in the presence of Fe(III), which forms stable complexes with Sb(III), holding them in a solution; thus, it inhibits antimony sorption.

The removal of both antimony anions, Sb(III) and Sb(V), from aqueous solutions was evaluated using seven kinds of Zr-MOFs with different aperture sizes and organic linkers containing various functional groups (-OH, -NH_2_, and -SO_3_H [84]. Zr-bound hydroxides in Zr-MOFs can simultaneously remove both Sb(III) and Sb(V) via an anion exchange mechanism. The NU-1000 framework with pyridine-2,4,6-triyl)tribenzaldehyde (TBAPy) linker (Figure 6A) exhibited the highest sorption capacity for Sb(III) (136.97 mg/g) and Sb(V) (287.88 mg/g). The experimental results were fit well to the Langmuir model. Sb(III) sorption was nearly constant in the pH range 2 to 11, while Sb(V) sorption was favorable at acidic pH and decreased at pH 11 (Figure 6B). Outer-sphere complexation based on an ion exchange process was suggested as a mechanism for the uptake of Sb(III) and Sb(V) ions by NU-1000 (Figure 6C). Almost all of the retained antimony can be eluted from NU-1000 by treating with 0.5 M HCl solution. 

### 2.6. Carbon Nanostructures Modified with Porphyrins

Porphyrins also serve as attractive bridging ligands in carbon nanostructures due to their rigid molecular structures, variable peripheral substituents, and large physical dimensions [85,86,87,88]. The modification of carbon nanostructures with porphyrins can be achieved through both covalent and non-covalent functionalization methods. Figure 7 illustrates the preparation of graphite oxide with covalently and non-covalently linked porphyrin [87].

A common covalent approach involves forming an amide bond between the carboxyl groups of carbon adsorbents and the amine groups in porphyrin molecules. Although this method is time-consuming, requiring multiple synthesis steps and yielding a relatively low degree of functionalization, it results in a more stable nanocomposite. In contrast, non-covalent functionalization provides a simpler synthesis route with high yields, relying on fundamental molecular interactions such as π-π interactions, van der Waals forces, hydrogen bonding, and electrostatic forces between the two components.

Behbahani et al. synthesized a biocompatible nanostructured carbon material by fructose hydrothermal condensation, followed by modification with amine groups [85]. Functionalization with tetraphenylporphyrin (TPPH2) was performed according to the general reaction of amine and carboxylic groups in porphyrin ligand. This sorbent exhibits a BET surface area of 289 m^2^/g. The quantitative removals for heavy metal ions (Cd(II), Ni(II), Cu(II), Fe(III)) were achieved at pH 6–7. The sorption kinetics decreased in the order Fe(III) > Ni(II) > Cd(II), Cu(II). Elution of sorbed metal ions and regeneration of carbon material were performed with 2 M HCl solution. There was no significant decrease in removal efficiency after 15 sorption/desorption cycles. The results showed high removal efficiency of this sorbent in real samples such as river and sea waters, and planting company wastewater.

Zhang et al. reported a novel ternary material of porous carbon nanofibers loaded with TCPP-Cu MOF and TiO_2_ nanoparticles [89]. This material was used for the colorimetric detection (LOD of 24.1 nM) and photoreduction of Cr(VI) in water samples.

Grafting covalent organic frameworks with a large number of specific uranium-binding groups and redox-active sites on the surface of carbon nanotubes can effectively enhance the sorption of uranium ions [90]. CNTs have good electron transport capacity and π-conjugation properties; thus, electrons can be effectively transferred from CNTs to COFs, improving the removal efficiency. CNT/COF-OH material had a sorption capacity of 518.2 mg/g. This sorption capacity was increased by 390.7%, 54.6%, and 84.5%, compared with those of single CNTs, COF-OH, and mixed CNT + COF-OH, respectively. The removal rate of U(VI) reached 96.7% for the rare earth tailings wastewater.

## 3. Comparison of Porphyrin-Based Sorbents for the Enrichment and Removal of Metal Ions

Different types of porphyrin-based sorbents have been developed for the preconcentration and removal of metal ions. The key factors influencing their performance, such as sample pH, sorbent dosage, sample volume, and the effect of coexisting ions, have been studied and optimized. Table 1 summarizes the latest data.

**Table 1 molecules-30-02238-t001:** Sorption of metal ions onto porphyrin-based materials.

Metal Ions	Sorbent	Sorption Conditions			q_max_ (mg/g)	Desorption/ Regeneration	Ref.
		Dose (mg)	pH	Time			
Au(III)	TNPPH_2_-phenazine COP	10	2	48 h	1354	0.1 M Th + 0.1 M H_2_SO_4_, 6 h	[60]
	Imi-P-POP	2	6	48 h	1543	10% Th + 5% HCl	[62]
	TCPP-MOP in dark +light glow +acetic acid	1	3	4 h	885 2613 4996	*na*	[80]
Au(I), Au(III)	(Fe(III)-TCPP)_n_-MOF	2	3	5 min	Au(I)-2026 Au(III)-2296	36% HNO_3_ + 4% HCl	[81]
Cd(II)	TAPP-COF	40	8	60 min	181	0.1 M HCl	[64]
	PS@TAPP-COF	40	8	35 min	166	0.1 M HCl, 0.1 M EDTA	
	CF@TAPP-Tp-COF	80	8	15 min	75.2	0.1 M HNO_3_	[65]
Cu(II)	SiO_2_@4TF_5_PP	10	6	15 min	184.15	6 M HCl	[51]
Cr(VI)	TPP-COF	2	na	4 h	293	0.3 M NaBr, 5 h	[66]
	T4MPP-CMP	0.15 mg/L	2	17 min	339.17	0.4 M NaOH, 2 h	[67]
	Imi-TPP-COF	50	2	5 min	373.14	na	[68]
	TCPP-COF	15	2	5 h	246.76	na	[70]
	Zn-TCPP-COF	15	2	60 min	273.09	na	
Hg(II)	Thiophene-P-POP	na	7	30 min	1049	na	[71]
	DA-P-POP	5	6	5 min	384.6	1 M HCl	[69]
Pb(II)	SiO_2_@TCPP	10	6	25 min	182.16	6 M HCl	[50]
	FeO@SiO_2_@TCPP		6	25 min	798.34	na	[54]
Sb(V)	Zr-TCPP-MOF		3	180 min	250.22	na	[82]
Sb(III)	Zr-TCPP-MOF	0.7 g/L	2	3 h	175.17	0.5 M HCl	[83]
Sb(III), Sb(V)	Zr-TCPP-MOF	8	2	9 min	Sb(III)-136.97 Sb(V)-287.88	na	[84]
U(VI)	CNT@COF-OH TPP-AO-HCP	5 5	5 6	12 h 6 h	518.2 110	na 2 M HCl	[90,91]
V(V)	SiO_2_@TCPP	10	6–8	60 min	35.0	2 M HNO_3_	[52]

COF—covalent organic framework; CMPs—conjugated microporous polymers; P-POP—porphyrin-based porous organic polymer; P-MOFs—porphyrin-based metal–organic framework; TNPPH_2_—5,10,15,20-tetrakis(4-nitrophenyl)porphyrin; Imi-P-POPs—imidazolium/porphyrin functionalized POPs; TAPP—5,10,15,20-tetrakis(4-aminophenyl)porphyrin; Tp—1,3,5-triformylphloro- glucinol; TPP—5,10,15,20-tetrakis(4-pyridyl) porphyrin; T4MPP—5,10,15,20-tetrakis(4-methoxyphenyl)porphyrin; TF_5_PP—5,10,15,20-tetrakis(pentafluorophenyl)porphyrin; Th—thiourea; PS—polyurethane sponge; CF—carbon fiber; DA—dialdehyde; HPCs—hyper-cross-linked polymers; AO—amidooxime group, na—not available.

Due to possible electrostatic interaction, the removal process is more efficient when the sample pH is higher than the point of zero charge of a given porphyrin-based material. Sorption efficiency generally increases with a higher sorbent dosage and longer extraction time until equilibrium is reached. Elevated temperatures typically increase the diffusion rate of metal ions towards the sorbent surface, which results in an enhanced sorption process. Materials with the shortest equilibrium time are the best choice as practical and applicable sorbents [68,69,81].

Sorption capacity (q_max_), the most extensively studied property of sorbents, reflects their ability to uptake specific metal species and can be used for comparison with other materials. There is no doubt that the highest q_max_ values have been reported when metal ions (such as Au, Cr, Sb, or U) are retained by a sorbent and subsequently undergo reduction to their metallic form (Table 1). Thus, the adsorptive-reductive mechanism was considered to play the key role in the uptake of these metal ions. Several studies highlight the important role of sulfur-containing functional groups, such as -S-, -SH, and -C-S-, in gold recovery. According to Qin et al. [92], a sulfhydryl-functionalized, zirconium-based MOF exhibited a high recovery capacity for Au(III) (1021 mg/g), while an MOF (UiO-66-NCS) combined with amidinothiourea achieved a value of 903.02 mg/g [93]. A sorbent prepared by covalently bonding 4-amino-3-hydrazino-5-mercapto-1,2,4-triazole onto graphene oxide demonstrated excellent performance not only for Hg(II) (q_max_ of 1091.1 mg/g), but also for Cr(VI) (734.2 mg/g), Cu(II) (168.9 mg/g), Pb(II) (103.4 mg/g), and Cd(II) (101.0 mg/g) [94].

Several different adsorption isotherm models have been used to provide valuable insights into the adsorption process, including the type of sorption (physisorption or chemisorption), the form of the adsorbed layer (monolayer or multilayer), and the nature of the adsorbent surface (homogeneous or heterogeneous) [93]. Generally speaking, the process of removal of metal ions by most porphyrin-based sorbents follows the Langmuir model, which indicates that this process is monolayer. A rapid increase with increasing concentration of metal ions, followed by a plateau, indicates chemisorption. Only in a few studies did the Freundlich model better describe multilayer adsorption on heterogeneous surfaces [70,71,81]. This statement was supported by the higher values of the correlation coefficient (R^2^) obtained from the fitted curves of the nonlinear Freundlich model.

### 3.1. Thermodynamic and Kinetic Parameters

The thermodynamic parameters provide specific insights into the energy changes occurring during sorption. The positive values of ∆H° indicate that the sorption process of metal ions is endothermic. The positive ∆S° values are typically associated with increased disorder at the solid–solution interface. The increasingly negative values of ∆G° with rising temperature demonstrate that the sorption process is spontaneous, thermodynamically favorable, and more efficient at higher temperatures. Based on all the cited studies in which these thermodynamic parameters were reported [50,51,53,62,67,68,70,91], it can be concluded that the removal and enrichment of the studied metal ions using a porphyrin-based sorbent is an endothermic, spontaneous, and thermodynamically favorable process.

The mass transfer of metal ions to sorbents involves the diffusion of ions from the solution to the surface of the material (film diffusion), diffusion within the solid matrix (particle diffusion), and the chemical interaction between the ions and functional groups. Identifying the rate-limiting step is crucial for improving sorption performance. For instance, if internal diffusion is the limiting factor, increasing the porosity of the adsorbent can accelerate the process [95].

Common kinetic models include the pseudo-first-order, pseudo-second-order, intraparticle diffusion, and Elovich models [14]. The selection of the appropriate model is typically based on the coefficient of determination (R^2^) obtained from fitting experimental data. A review of the literature indicates that the pseudo-second-order kinetic model most consistently fits the experimental data. The majority of calculated rate constants (k_2_) for this model fall within the range of (1.1–6.1) × 10^−^^3^ g/mg·min [50,51,65,66,67,71], although significantly lower values, in the range of 0.3–0.4 × 10^−^^3^ g/mg·min, have also been reported [62,82,92].

Furthermore, the adsorption behavior of metal ions on porphyrin-based materials, analyzed using the intraparticle diffusion model, suggests that the process occurs in two [83] or three [68] distinct stages, with none being dominant. Initially, external surface adsorption likely occurs rapidly, followed by a moderate internal diffusion stage, and finally, a slow approach to equilibrium.

### 3.2. Selectivity

Porphyrin-based sorbents can effectively interact with various metal ions, even when there is a size mismatch between the macrocyclic cavity and the ionic radius of the target ion. When a metal ion is too large to fit into the porphyrin cavity, it typically positions itself above the plane of the pyrrolic nitrogen atoms, forming a sitting-atop (SAT) complex [96]. Although complexation reactions generally favor a planar structure, the geometry of porphyrins can be distorted by the nature of the central metal ion and the presence of peripheral substituents [97]. These distortions can affect the porphyrin’s basicity, redox potential, reactivity, and coordination behavior toward metal ions. Differences in selectivity can be illustrated by the comparison of the sorption capacity values for various metal ions. The selectivity order for SiO_2_@TF5PP was Pb(II) > Cu(II) > Cd(II) > Zn(II) [50]; for SiO_2_@4TF5PP: Cu(II) > Pb(II) > Zn(II) > Cd(II) [51]; and for SiO_2_@NTPP Pb(II) > Zn(II) > Cd(II) > Cu(II) [53]. Apart from the variations in metal ionic radii, the observed differences suggest that other mechanisms may influence the sorbents’ selectivity. Most likely, the presence of substituents and the modification of the electronic structure of the porphyrin core affect selectivity by enabling different structural adaptations.

Generally, the extraction efficiency for a target metal ion using porphyrin-based sorbents decreased when the multicomponent mixture was tested, and an example is shown in Figure 8 [51].

Smaller or larger interferences from competitive metal ions were also observed for other porphyrin-based materials. Compared with PS@2,5-Tp-COF, PS@2,3-Tph-COF showed higher resistance to interferences [64]. Above 99% of Cr(VI) was selectively removed by Imi-TPP-COF, while the removal efficiency for Cd, Hg, Ni, and Co was below 5% [68]. The removal rate of the Imi-P-POP network for Au(III) under the possible interferences of multiple competing metal ions was about 80%, while the removal rates for Hg(II) were nearly 40%, and 30% for Cr(III) [62]. On the other side, Imi-P-POP exhibited a high gold removal rate (89%) in the leaching solution of an authentic e-waste, and the removal rates for the matrix metal ions, such as Cr(III), Zn(II), Co(II), Ti (IV), Cu(II), Ag(I), Hg(II), and Mn(II) were 0%, 1.6%, 2.2%, 9.1%, 10.6%, 13.4%, 22.8%, 35.5%, and 40.0%, respectively.

It should be noted that the presence of other ions typical for natural water samples (Na^+^, K^+^, Ca^2^^+^, Mg^2^^+^, NH_4_^+^, SO_4_^2^^−^, NO_3_^−^, HCO_3_^−^, PO_4_^3^^−^) did not interfere with the sorption of the target metals.

### 3.3. Desorption of Metal Ions and Regeneration/Stability of Sorbents

After the sorption process, the retained metal ions must be released from the used material for recovery and further use. In several papers, diluted mineral acid solutions were used [52,64,69,83,85]. More drastic chemical conditions in the desorption step, like very concentrated hydrochloric or nitric acids, were also proposed [50,81]. The stability of the organic groups on the solid surface was confirmed with no distinct changes. The reusability of the sorbents, which is of prime importance concerning economic efficiency, was evaluated via consecutive adsorption and desorption cycles, usually five.

Li et al. regenerated the TPP-COF network used for the removal of Cr(VI) by soaking in NaBr solution (13 g/L) for 5 h [66]. An excess of bromide anions as a competing species can effectively displace Cr(VI) by binding to the COF more strongly. The regenerated material displayed almost identical FTIR and ^13^CNMR spectra at the beginning, indicating preservation of its structure, but its sorption percentage dropped to about 78% after five cycles.

Good efficiencies of gold desorption were achieved using acids (HCl, HNO_3_, H_2_SO_4_), their mixtures, and acidic thiourea solutions [60,61,62,63,80,81]. The rate of desorption was higher when the temperature was raised to 50 °C [61]. However, it was found that the characteristic peaks of the porphyrin structure in FTIR spectra of COP-180 (produced with TNPPH_2_ as a monomer) remained after gold desorption with thiourea, while after application of HCl-HNO_3_ solution, most of these peaks diminished or even disappeared, indicating disruption of the structure [61]. Son et al. found that a small (~5%) portion of gold is irreversibly retained on the porphyrin structure, and it remained virtually unchanged after multiple regeneration cycles. The decrease in micropore size and pore volume, slight decomposition of sorbent, and non-complete retention after desorption were also observed for the Zr-TCPP-MOF network evaluated for Sb(III) sorption [83].

Nuyen et al. examined the stability of porous porphyrin–phenazine polymers used for gold recovery from e-waste [60]. The desorption/regeneration process was conducted for 6 h at 80 °C. After only two cycles, this material lost about 15% of its maximum sorption capacity. Thus, it was calculated that its single use would be profitable if the gold price remained higher than USD 39 per gram. The gold recovery process of sorption and desorption using TNP-COP was also evaluated from an economic point of view [61]. In total, the cost of chemical reagents for the polymer synthesis and the desorption process was estimated to be USD 5.26 per gram of COP. The cost of energy consumed in both processes was calculated to be USD 0.13 to treat 1 g of this material. Based on the assumption that the average gold desorption efficiency in a single cycle is 94%, to yield one gram of this metal would cost USD 28.7.

### 3.4. Comparison of Porphyrin-Based Sorbents with Other Types of Sorbents

The functional properties of sorbents are defined by parameters such as sorption capacity, kinetics, and selectivity. A critical comparison of these parameters enables the assessment of the potential of porphyrin-based sorbents. For a general evaluation of the potential of porphyrin sorbents, a comparison can be made with Chelex-100, a widely used and commercially available sorbent. Reported sorption capacities for Chelex-100 include 15.55 mg/g for Cr(III) [98], 110 mg/g for Cd, and 22.8 mg/g for Pb [99]. A comparison of these values with the data presented in Table 1 demonstrates a significant advantage for porphyrin-based sorbents.

However, comparisons with the best-performing sorbents indicate that materials with similar sorption capabilities may emerge, and factors such as differences in adsorption capacity, sorption kinetics, and selectivity for heavy metal ions can significantly influence the overall evaluation of porphyrin-based sorbents. Although such comparisons are present in original research, they most often focus exclusively on adsorption capacity, with limited consideration given to kinetics or selectivity. Abiad et al. [51] presented a comparison of Si3PyS and Si4PyS performances vs. recently reported Cu sorbents, where porphyrin sorbents reached an adsorption capacity of 176–184 mg/g. Next in comparison, ketoenol-bipyridine has 131 mg/g, and the median value of results for the other 23 compared sorbents was 32.15 mg/g. Similar analyses performed for antimony [84], gold [81], cadmium [64], mercury [71], or multielement experiments [50,53] confirm comparable or better values of adsorption capacity of porphyrin-based materials in comparison to leading solutions for individual elements. For a more comprehensive assessment of the potential of porphyrin networks, Table 2 presents a comparison of other performance parameters, such as kinetics, durability and selectivity for competitive non-porphyrin sorbents.

The comparison highlights several specific properties of porphyrin-based sorbents. Owing to their photocatalytic activity, these sorbents exhibit high selectivity and sorption capacity for metal ions, particularly when redox equilibria facilitate the sorption process. In some cases, the sorption is so stable that clusters of reduced metals form within the sorbent structure. This phenomenon may contribute to a decrease in sorption capacity after regeneration and reduce reusability. When coordination interactions dominate, a distinct type of group selectivity emerges: metals as Cd, Cu, Zn, Pb are efficiently trapped, while metals commonly present as natural components (Ca, Mg) in environmental samples exhibit very low sorption. This characteristic makes some porphyrin sorbents particularly suitable for the selective removal of toxic metals from the environment, but the removal of metal ions from multicomponent systems remains poorly understood, affecting the control of selectivity and efficiency in the presence of competing ions.

**Table 2 molecules-30-02238-t002:** Comparison of porphyrin-based sorbents (*top row for each element*) with other state-of-the-art sorbents.

Metal Ions	Sorbent	q_max_ (mg/g)	Equilibrium Time	No. of Cycles	Selectivity	Ref.
Au(III)	*TNPPH_2_-phenazine COP*	*1354*	*30 min*	*>3*	>*93.4% of Au*	[60]
	Zr UiO-66-NH_2_ MOF	650	12.2 h	>5	Selective vs. Co, Ni	[100]
	Fe-BTC/PpPDA MOF	934	<2 min	>10	>99.9% of Au	[101]
Cd(II)	*PS@TAPP COF*	*166*	*40 min*	*>5*	>*85% of Cd*	[64]
	PANH	156	180 min	>3	>70% of Cd	[102]
	N-riched COF	396	40 min	na	na	[103]
Cu(II)	*SiO_2_@4TF_5_PP* SiO_2_@NP2 SiO_2_@NNN	*184.15* 131.82 121.6	*10 min* 25 min 20 min	*>5* na >5	Co-sorption of Cd, Pb, Zn Co-sorption of Cd, Pb, Zn >80% of Cu	[51,104,105]
Cr(VI)	*Imi-TPP-COF*	*373.14*	*40 min*	>*5*	>*90% of Cr*	[68]
	PTAPDAC	273.17	60 min	na	na	[106]
	CS-PVA hydrogel	320	200 min	3	na	[107]
	CCGP	179.2	30 min	3	na	[108]
Hg(II)	*Thiophene-P-POP*	*1049*	*30 min*	*5*	*Significantly selective*	[71]
	S-SH COF S-CX4P	1350 1686	10 min 5 min	na 4	Co-sorption of Cu, Pb Co-sorption of Cu, Zn, Mg, Ca	[109,110]
Sb(V)	*Zr-TCPP MOF*	*250.22*	*100 min*	*5*	*Strong negative impact of SO_4_^2−^*	[82]
	RGO@Mn_3_O_4_	105.50	25 min	na	Negative impact of PO_4_^3−^	[111]
	TiO_2_	156	60 min	na	Negative impact of PO_4_^3−^ and SO_4_^2−^	[112]

na—not available; BTC—1,3,5-benzenetricarboxylate; PpPDA poly-para-phenylenediamine; PANH—hydrolyzed beads of polyacrylonitrile; SiO_2_@NP2—ß-ketoenol-bipyridin; PTAPDAC—poly(N^1^,N^1^,N^3^,N^3^-tetraallylpropane-1,3-diaminium chloride; CS-PVA—chitosan/polyvinyl alcohol; CCGP—cross-linked chitosan grafted–polyaniline composite; S-SH COF—vinyl-functionalized with thiol mesoporous COF; S-CX4P—thioether-crown-rich calix[4]arene porous polymer.

The comparison highlights several specific properties of porphyrin-based sorbents. Owing to their photocatalytic activity, these sorbents exhibit high selectivity and sorption capacity for metal ions, particularly when redox equilibria facilitate the sorption process. In some cases, the sorption is so stable that clusters of reduced metals form within the sorbent structure. This phenomenon may contribute to a decrease in sorption capacity after regeneration and reduce reusability. When coordination interactions dominate, a distinct type of group selectivity emerges: metals as Cd, Cu, Zn, Pb are efficiently trapped, while metals commonly present as natural components (Ca, Mg) in environmental samples exhibit very low sorption. This characteristic makes some porphyrin sorbents particularly suitable for the selective removal of toxic metals from the environment, but sorption in multicomponent systems remains poorly understood, affecting the control of selectivity and efficiency in the presence of competing ions.

## 4. Summary

The structural versatility of porphyrins enables their application across a wide spectrum of fields. Porphyrin-based compounds serve as attractive ligands, allowing for efficient and selective sorption of metal ions. One major area of application involves the modification of classical adsorbents. Moreover, the potential to develop new materials solely based on porphyrin frameworks enables multifunctional roles beyond structural support. Their chemical structure is readily tunable—via functional group incorporation or metal complexation—to optimize selectivity and binding affinity for target metal ions. This versatility, combined with their adsorption, redox, and photochemical properties, offers a pathway toward the development of a novel class of functional two- and three-dimensional materials.

Porphyrin-based sorbents exhibit high affinity for certain metals as an advantage common to the entire class, but the variety of structures and material types available offers distinct advantages. Porphyrin–silica hybrids benefit from mechanical and thermal stability, tunable selectivity, and ease of regeneration, showing high efficiency, particularly for Pb(II) [50] and Cu(II) [51]. However, their sorption capacities are lower than advanced porous materials. Porphyrin-immobilized resins exhibit moderate capacity, but are limited by physical retention mechanisms (e.g., π–π interactions), especially in non-ionic resins. Porphyrin-based porous organic polymers (P-POPs) offer superior adsorption (up to 1543 mg/g for Au(III) [62] and 1049 mg/g for Hg(II)) [71], photoactive behavior, and tunable functionality, although their synthesis can be complex, and light-dependence may limit practicality. Porphyrinic MOFs combine structural rigidity with exceptionally high capacities (up to 4946 mg/g for Au(III) [80]), fast kinetics, and photoredox properties. Their main limitations lie in lower chemical stability and sensitivity to competing ions or pH shifts [82]. Carbon nanostructures modified with porphyrins offer high surface area and excellent reusability, though functionalization is labor-intensive. Overall, P-POPs and P-MOFs provide the highest sorption capacities and functional diversity, while silica and resin-based systems offer simplicity, recyclability, and environmental stability.

The applications discussed herein—such as electronic waste recycling, battery construction, and the selective removal of toxic chromium(VI) from aqueous environments—demonstrate the synergistic use of porphyrins’ chemical and structural properties. However, current research also reveals several gaps and challenges that must be addressed to enable practical implementation.

*Applicability*: Most investigations have been conducted under controlled and simplified laboratory conditions. However, data on the performance of porphyrin-based sorbents in complex, real-world matrices, such as industrial wastewater, mining effluents, or seawater, remain limited. This gap is further compounded by an insufficient understanding of competitive binding dynamics in multicomponent systems. Improving the selectivity and reusability of these materials under such conditions constitutes a critical step toward practical deployment. Moreover, economic considerations, despite encouraging preliminary findings, are rarely addressed in the current literature.

*Environmental Impact and Toxicology*: The environmental implications of porphyrins span both their synthesis and application. The synthesis of porphyrin-based materials commonly involves pyrrole and aldehydes, both potentially toxic reagents, and also several hazardous solvents, like chloroform, dichloromethane, and dimethylformamide. These methods often yield low product quantities and generate significant byproducts. During application, environmental risks arise from the incorporation of metal ions into porphyrin structures, potentially converting them into hazardous waste. Regeneration of these materials is not always quantitative, leading to poorly characterized metalloporphyrin waste. Furthermore, residual porphyrins, functional moieties, or counterions may pose ecological risks. The intrinsic photoactivity of porphyrins can further complicate their environmental behavior, potentially altering their persistence or inducing the generation of reactive oxygen species (ROS) upon light exposure.

*Continuous Improvement:* Despite considerable progress, porphyrin-based sorbents often exhibit slower uptake rates or lower capacities compared to current high-performance materials. Long-term stability and reusability under harsh environmental conditions, such as pH fluctuations, oxidative stress, or the presence of aggressive chemicals, remain key concerns. Moreover, selective sorption in multicomponent systems is not yet fully understood, and strategies to enhance selectivity amid competing ions are still under development. The toxicological profile of synthetic porphyrins, particularly post-metal-binding, also requires further elucidation.

Addressing these challenges is essential for transitioning porphyrin-based sorbents from laboratory research to field-ready technologies. Continuous advancements in synthesis and characterization techniques will be vital in expanding their application potential and in meeting the demands of practical environmental and industrial use.

## Figures and Tables

**Figure 1 molecules-30-02238-f001:**
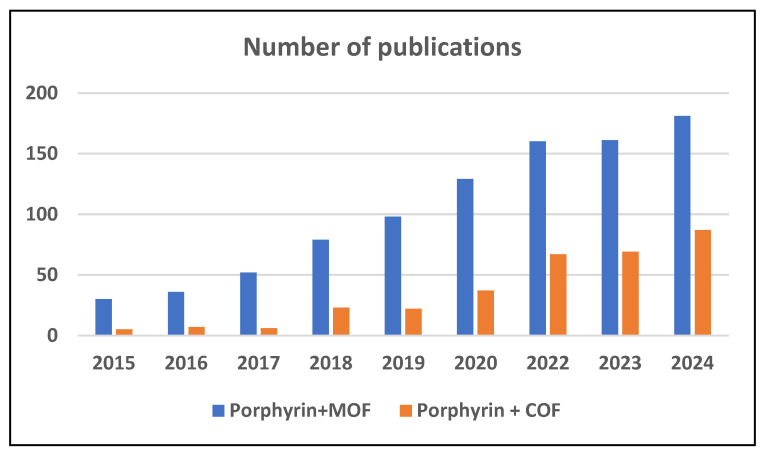
The number of publications on porphyrin-based sorbents (Scopus, keywords: “Porphyrin” AND “MOF”; “Porphyrin” AND “COF” as of 1 May 2025 in period 2015–2024).

**Figure 2 molecules-30-02238-f002:**
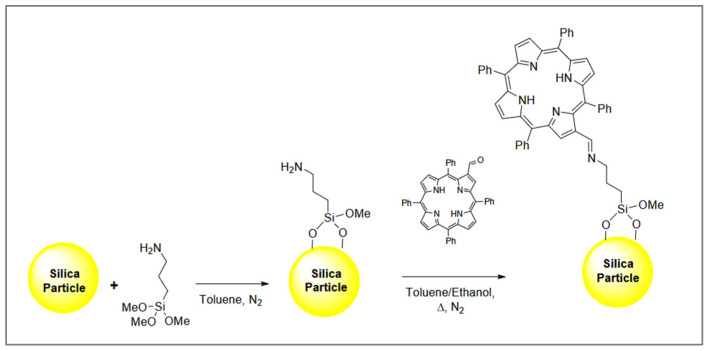
The scheme for the preparation of 2-formyl-5,10,15–20-tetraphenylporphyrin grafted silica. Reproduced with permission from Ref. [51]. Copyright Elsevier 2019.

**Figure 3 molecules-30-02238-f003:**
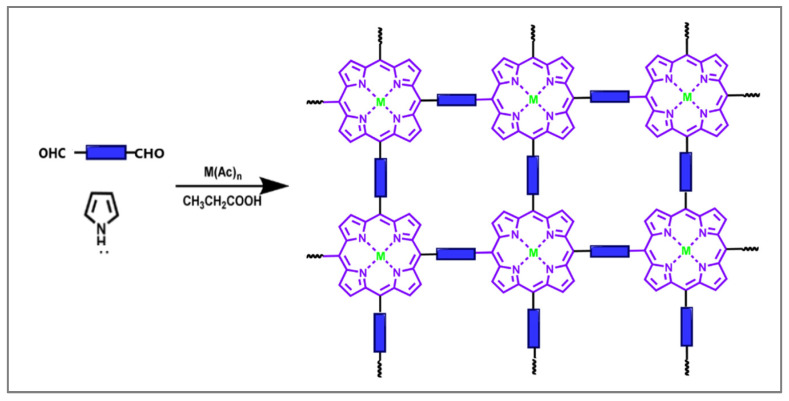
Schematic route for the synthesis of porphyrin-based porous organic polymers by one-pot synthesis [40]. Reprinted under the terms of the CCA 3.0 license from reference [40]. Copyright Royal Society of Chemistry 2024.

**Figure 5 molecules-30-02238-f005:**
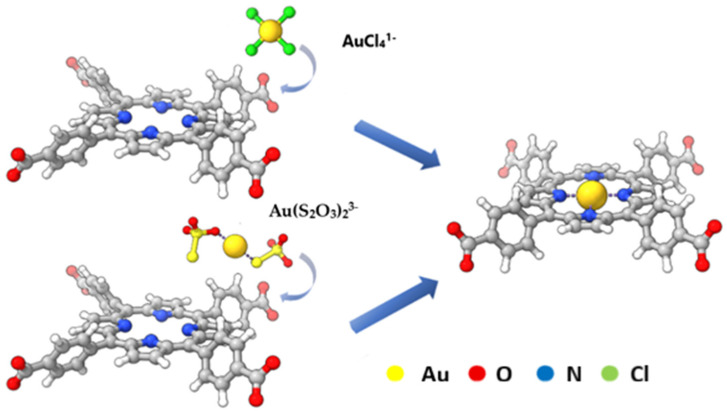
Schematic diagram of Au(III) and Au(I) sorption on the porphyrin part of (Fe-TCPP)_n_-MOF. Reproduced with permission from reference [81]. Copyright Elsevier 2025.

**Figure 6 molecules-30-02238-f006:**
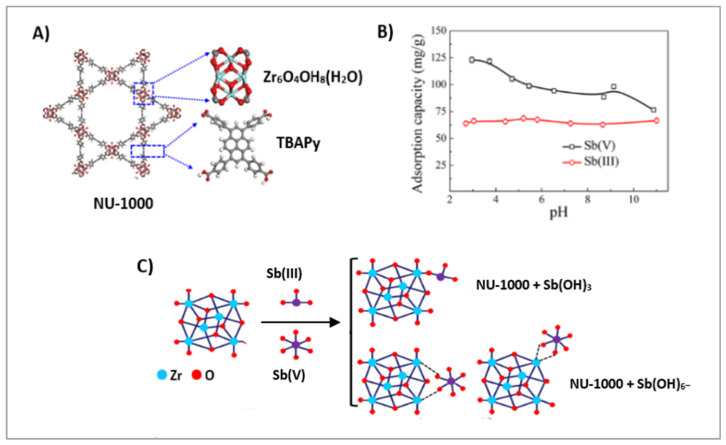
Structure of NU-1000 Zr-MOF (**A**). Influence of pH on Sb(III) and Sb(V) sorption on NU-1000 (**B**). Potential binding modes of Sb(III) and Sb(V) (**C**). Reproduced with permission from reference [84]. Copyright American Chemical Society 2017.

**Figure 7 molecules-30-02238-f007:**
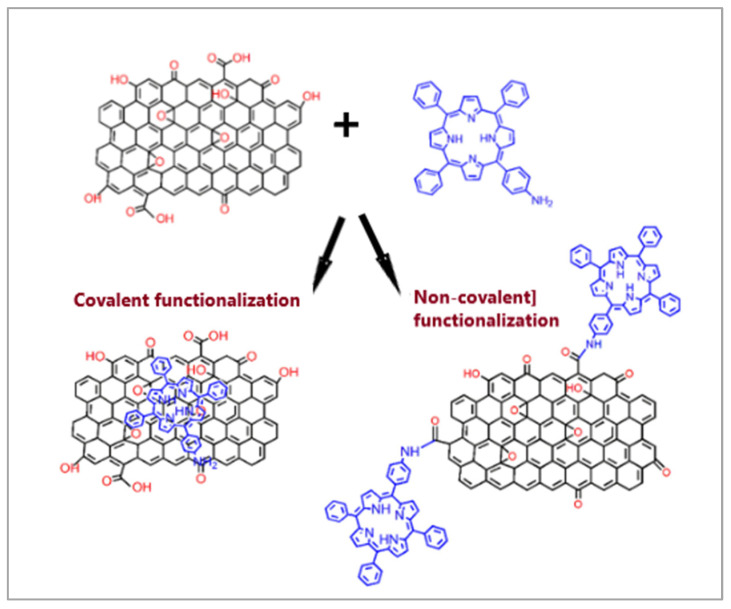
Scheme for the preparation of graphite oxide with covalently and non-covalently linked porphyrin. Reprinted under the terms of the CCA 3.0 license from reference [87]. Copyright Royal Society of Chemistry.

**Figure 8 molecules-30-02238-f008:**
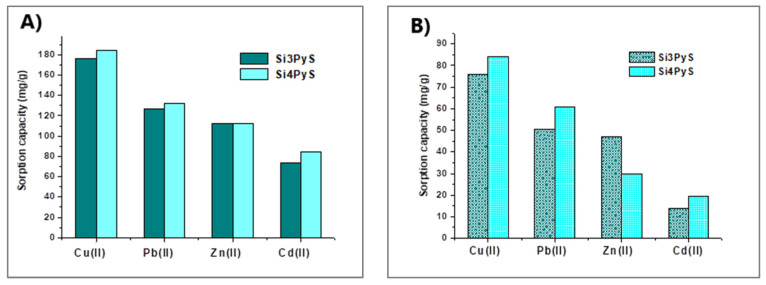
Sorption capacity of some metal ions on SiO_2_@TF_5_PP with three (Si3PyS) and four (Si4PyS) mercaptopyridyl units. (**A**) determined in single metal ions in a solution; (**B**) determined in the quaternary mixture. Reprinted under the terms of the CCA 3.0 license from reference [51]. Copyright Elsevier 2023.

## Data Availability

No new data were created or analyzed in this study. Data sharing is not applicable to this article.

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
