# Peer review of "Porphyrin-Based Sorbents for the Enrichment and Removal of Metal Ions"

_molecules, 2025, doi:10.3390/molecules30102238_

Round 1

Reviewer 1 Report

Comments and Suggestions for Authors

The article is a comprehensive review of porphyrin-based materials used for metal ions sorption. It discusses various forms of support and particularly deals with sorption capacities and mechanisms of porphyrin derivatives in applications ranging from environmental remediation to analytical chemistry.

The manuscript is really well-written, and I suggest its publication after considering some points that might improve the overall discussion:

  1. the review illustrates the performance of porphyrin-based materials but does not show a comparison with common state-of-the-art sorbents. A quantitative comparison in terms of sorption capacity, kinetics, and selectivity could be added;
  2. some reported materials have been tested on real wastewater, but long-term stability, regeneration over many cycles, and potential industrial scalability are not fully addressed. Could any of these aspects be considered?
  3. What about the safety and possible environmental concerns about synthetic porphyrins and some functionalization steps? Aspects related to possible release of dangerous functional groups should be addressed.

Reviewer 2 Report

Comments and Suggestions for Authors

In this article, the authors thoroughly investigate the various types of porphyrin-based materials, their production processes, and their application as sorbents for the enrichment and removal of metal ions. The manuscript is comprehensive and well-organized. Additionally, the subject of the paper is timely and may be of interest to the journal's readers. The review appears appropriate for the special issue; however, minor revisions are necessary prior to acceptance.

  1. To improve the manuscript’s structure and clarity, a paragraph describing the research methodology should be added at the end of the Introduction section.
  2. The authors should include the publication trend of scientific articles on porphyrin-based materials used as sorbents for metal ions over the past ten years.
  3. It is recommended that a more thorough analysis of the characterization of porphyrin-based materials be included as a separate section. Furthermore, a comparative summary of the advantages and disadvantages of the four discussed porphyrin-based material groups should be presented in table form.
  4. Please add a few sentences at the beginning of each paragraph about metals, their characteristics, where they are found, and their effects on health.
  5. A separate section should be added as "Regeneration of porphyrin-based materials" for this review paper.
  6. Please support the sentences with appropriate references (e.g., Line 42-43, 59-60, 232-233).

Reviewer 3 Report

Comments and Suggestions for Authors

The manuscript presents a comprehensive review of the development and application of porphyrin-based sorbents for the enrichment and removal of metal ions from aqueous systems. The review consolidates knowledge on a diverse range of porphyrin-based materials (e.g., porous polymers, MOFs, COFs, nanocomposites), highlighting their physicochemical properties, sorption performance, and applications. However, the current form lacks critical depth, organization, and editorial polish expected of a publishable review. A major revision addressing the following issues could transform this into a valuable contribution to the field.

  1. In lines 73–76 of the Introduction, the authors state that a distinguishing feature of this review, compared to other review articles on porphyrin-based materials, is its focus on their role in analytical chemistry, particularly in metal enrichment prior to quantification and in speciation analysis. However, this specific focus does not appear to be sufficiently emphasized in the subsequent sections. The authors are encouraged to revise the manuscript to better reflect this stated objective.
  2. In Section 2, the authors introduce six major categories of porphyrin-based sorbents. However, the discussion primarily consists of a summary of existing studies related to the synthesis of these materials, without providing a critical comparison regarding the types of metal ions each sorbent is most suitable for, the underlying sorption mechanisms, the differences among the sorbent types, and their respective limitations in practical applications. The authors are encouraged to expand this section by incorporating a more analytical and comparative evaluation of these aspects.
  3. In lines 129, 152, and 183, the subsection headings are all labeled as “2.1,” which appears to be a formatting error. The authors are kindly requested to correct the numbering of these subsections to ensure consistency and clarity in the manuscript structure.
  4. Overall, the manuscript lacks a clear and coherent thematic progression. For instance, while various porphyrin-based sorbents are categorized and discussed in Section 2, Section 3 shifts to a classification based on metal ions, leading to repeated transitions between different sorbent types. Given that the main focus of this review is on porphyrin-based sorbents, it would be more appropriate to organize the manuscript around the six major sorbent categories introduced in Section 2. Within each category, the discussion should follow a consistent structure: synthesis method → material characterization and properties → sorption mechanism → application examples (e.g., which metal ions can be adsorbed). This structure would provide greater clarity and allow the authors, in the concluding section, to more effectively carry out the critical comparison suggested in Comment 2
  5. In the final summary, the authors do not sufficiently address the existing research gaps or unresolved challenges in the development of porphyrin-based sorbents. It is recommended that the authors expand this section by discussing the following points in light of the current state of research: (1) The potential challenges these materials may face in transitioning toward commercial or large-scale applications; (2) Possible environmental concerns associated with their synthesis or regeneration processes; (3) Opportunities for further improvement and the future development potential of porphyrin-based sorbents.

Round 2

Reviewer 1 Report

Comments and Suggestions for Authors

The manuscript has been improved, and in my opinion it deserves to be published. 

Reviewer 3 Report

Comments and Suggestions for Authors

No more comments